# DNA barcoding for elasmobranch diversity assessment in Thailand: Its advantages and limitations

Jenjit Khudamrongsawat[1], Tassapon Krajangdara[2], Thadsin Panithanarak[3], Ratima Karuwancharoen[3], Wanlada Klangnurak[4], Pattarapon Promnun[5], Wansuk Senanan[6]*

1 Department of Biology, Faculty of Science, Mahidol University, Bangkok, Thailand, 2 Independent Researcher, Phuket, Thailand, 3 Institute of Marine Science, Burapha University, Chon Buri, Thailand, 4 Department of Animal Production Technology and Fishery, School of Agricultural Technology, King Mongkut's Institute of Technology Ladkrabang, Bangkok, Thailand, 5 Geneus Genetics Co., Ltd, Bangkok, Thailand, 6 Department of Aquatic Science, Faculty of Science, Burapha University, Chon Buri, Thailand

* wansuk@buu.ac.th, wansuks2@yahoo.com

## Abstract

The assessment of elasmobranch biodiversity in Thailand benefits greatly from the application of DNA barcoding, which helps mitigate the challenge posed by a shortage of expert taxonomists. Fragments of COI and ND2 mitochondrial DNA were examined, and the strengths and weaknesses of these two markers were compared. In this study, DNA products from 153 elasmobranch samples were amplifiable and revealed a total of 28 shark species and 32 batoid species. Many species could be confidently identified as their morphological characteristics aligned with DNA barcodes. However, several exceptions were recognized. The absence of reference sequences for rare species presented a challenge for species verification, and the misidentification of reference sequences, as well as changes in species names due to taxonomic revisions, added complexity when comparing DNA barcoding sequences. Conflicts between morphology and genetics were also observed. While intraspecific genetic variation based on both DNA barcodes generally indicated 0–2% variation, this metric could not always be used for species delimitation. This was particularly true for species displaying low genetic variation among closely related species and species where cryptic diversity remained hidden and yet to be uncovered. In such cases, the morphological characteristics of the samples served as the primary means of species identification. Despite these challenges, DNA barcoding remains an invaluable tool for biodiversity assessment, especially in light of the shortage of skilled experts, and for identification of products made from vulnerable species. However, it is essential to exercise caution and be aware of these complexities in its application.

**Data availability statement:** All relevant data are within the manuscript and its Supporting information files. We included all the Genbank accession numbers for the nucleotide sequences in the Supporting information file.

**Funding:** This project is funded by the National Research Council of Thailand (NRCT) under grant number N25A650485 (WS, JK, TK, TP, RT, WK). The funder had no role in study design, data collection and analysis, decision to publish, or preparation of the manuscript.

**Competing interests:** The authors have declared that no competing interests exist.

## Introduction

The Southeast Asian region (SEA) boasts a remarkable array of elasmobranch species in which the ongoing discovery of new, undescribed species is adding to our appreciation of the extraordinary diversity of this under-studied group [1,2]. To safeguard these taxa in the region, the Southeast Asian Fisheries Development Center (SEAFDEC) has been actively encouraging SEA countries to assess their elasmobranch diversity so as to better formulate international conservation policies. Among SEA nations, Thailand is one of chondrichthyan biodiversity hotspots. This is attributable to the country's expansive marine ecosystem covering nearly 3,000 kilometers of coastlines and spanning both the Indo-West Pacific region and the Indian Ocean. The geological dynamics and intricate evolutionary history further contribute to this wealth of biodiversity [3,4]. The latest records unveil over 180 species (across 12 orders and 43 families), including some apparently distinct, yet morphologically similar lineages that are yet to be confirmed as full species [5]. Evident disparities exist in elasmobranch diversity between the Gulf of Thailand and the Andaman Sea. The species composition within the Gulf of Thailand tends to closely resemble that of the western Pacific region, whereas the Andaman Sea species are more similar to those prevalent in the east Indian Ocean. Notably, the Andaman Sea boasts a higher number of elasmobranch species compared to the Gulf of Thailand. This discrepancy is further underscored by the presence of Holocephali species exclusively recorded in the Andaman Sea [5,6]. The escalating species count is a result of enhanced survey efforts and the integration of molecular tools for species identification. Despite ongoing surveys and the submission of DNA barcode information to the NCBI database and Barcode of Life Database (BOLD), Thailand's remarkable elasmobranch diversity remains poorly documented in the existing literature [7–10]. Available genetic information, though scant, poses many questions regarding taxonomic diversity, particularly the high levels of genetic differentiation that cannot be overlooked [11].

The evaluation of elasmobranch diversity hinges on the utilization of both morphological and molecular tools, with DNA barcoding gaining prominence, especially via the cytochrome c oxidase subunit I (COI) and the nicotinamide adenine dehydrogenase 2 (ND2) gene fragments [7,12–15]. DNA barcoding facilitates the differentiation of cryptic species that are difficult to distinguish based on appearance alone [16]. Even species with extensive and apparently continuous distributions may exhibit genetic variations that have implications for disparate evolutionary trajectories [17,18]. Specifically, ND2 has shown superiority in uncovering genetic variation in cryptic sharks and is strongly recommended for use in chondrichthyan DNA barcoding [7,19]. While DNA barcoding offers a powerful and expeditious tool for assessing biodiversity, it has proven challenging to establish clear criteria for genetic differentiation between species, particularly for non-taxonomists. In many instances, genetic variation within elasmobranchs falls within a 2% range at the intraspecific level [13,15,20–22]. However, exceptions do exist [13,15,19,23,24]. These strengths and weaknesses of DNA barcoding for biodiversity assessment have been previously acknowledged for various taxa [25–33]. Nonetheless, as Thailand has an urgent

need to improve the country's biodiversity assessment, the adoption of DNA barcoding is a worthwhile endeavor. Genetic insights into species identification have yielded significant advances in shaping the Thailand National Plan of Action for the Conservation and Management of Sharks (NPOA-Sharks, Thailand). This aligns with the nation's economic and development agenda in accordance with the United Nations' 2030 Sustainable Development Goals (SDGs).

The objectives of this research were to evaluate the extent of DNA barcoding application based on COI and ND2 fragments for both taxonomists and non-taxonomists in elasmobranch biodiversity assessment and to create DNA barcoding databases of elasmobranch diversity in Thailand. The outcomes hold the potential to provide valuable contributions to the regional conservation efforts by shedding light on Thailand's unique biological resources.

## Materials and methods

### Tissue sampling locations and collections

Permission for animal use was granted by MU-IACUC (permission MUSC62-025-489) and Burapha U-ethics (IACUC 002/2566). Samples were obtained from various sources, encompassing research vessels, fish markets, and fishing harbors situated in both the Gulf of Thailand and the Andaman Sea over the period from 2013 to 2023. These specimens were photographed and underwent subsequent species identification by T. Krajangdara following the guidance provided in Krajangdara et al. [5], Last et al. [34], Krajangdara [35], and Ebert et al. [36]. Instances where characteristics proved intricate or displayed unclear attributes were categorized as unidentified species, designated with "cf." or "sp." To definitively classify these cases, molecular analysis was performed. Muscle or fin clips were excised from the specimens and preserved in 95% ethanol, subsequently being frozen at -20°C at Burapha University, Mahidol University, King Mongkut's Institute of Technology Ladkrabang, and the Natural History Museum of Thailand. In cases involving sizeable, expensive, or live individuals, photographs were taken as records, with the focus being solely on acquiring tissue samples. To ensure the future viability of the research, selected samples were designated as voucher specimens and retained at these institutions: the Phuket Marine Biological Center, Burapha University, King Mongkut's Institute of Technology Ladkrabang, and Natural History Museum of Thailand.

### Molecular examination

DNA samples were extracted from more than 200 samples using a commercial extraction kit (QIAGEN—DNeasy kit). The quality of the extracted DNA samples was assessed by electrophoresis on a 1% agarose gel. Their concentrations were measured using a Nanodrop (NanoVue™). DNA barcoding was based on cytochrome c oxidase subunit I (COI) and NADH dehydrogenase subunit 2 (ND2), which were amplified following the protocols by Ward et al. [12] and Naylor et al. [7]. Degraded tissue samples, especially those obtained before 2020, could not be amplified using universal primers, and so new primers were designed for investigation. A list of primers used in this study appeared in Supporting Information (S1 Table). Polymerase chain reactions (PCR) were carried out with a total reaction volume of 25 ul using 5–7 ng of DNA template, 2 µl (0.2 mM) dNTP, 2.5 µl 10xPCR buffer, 1 µl (0.5 µM) of each primer, 1 µl (2 mM) $MgCl_2$, 0.05 µl (5 unit/µl) Taq DNA polymerase, and deionized water. PCR cycles were performed with an initial denaturation at 94°C for 3 min, followed by 35 cycles of 94°C for 50 s, annealing at 50°C for 2 min, extension at 72°C for 90 s, and a final extension at 72°C for 6 min. For samples posing difficulty in amplification, a touchdown PCR profile was performed including an initial denaturation at 94°C for 3 min, followed by 5 cycles of 94°C for 30 s, annealing at 48°C for 60 s, and extension at 72°C for 90 s, and then followed by 30 cycles of 94°C for 30 s, annealing at 50°C for 60 s, extension at 72°C for 90 s, and a final extension at 72°C for 10 min. PCR products were checked by electrophoresis using 1.5% agarose gel and purified using commercial Gel/PCR purification kit (QIAGEN—QIAquick PCR purification kit). Sequences were sent for Sanger sequencing using an ABI 3730 automated sequencer. All sequences were edited, then translated into amino acids using MEGA11 [37].

## Data analysis

Species were confirmed through a dual process involving both morphological identification and cross-referencing with reference sequences available in the National Center for Biotechnology Information (NCBI) and the Barcode of Life Database (BOLD system). Taxonomic name changes, particularly in the family Dasyatidae, were also recognized [38]. The differences in the generic names of species in the genera previously named as *Himantura* and *Dasyatis* were not treated as different from the newly assigned names. In instances where matches were inconclusive, a thorough assessment was conducted using supplementary information, notably morphology and geographic data. In cases of persistent uncertainty, the designation "cf." was attributed to those particular samples. The validation of species through molecular means relied on a similarity scoring system, demanding a minimum of 95% query coverage and a similarity level of 98% or higher. The 95% query coverage threshold was selected for its robustness and greater likelihood of yielding accurate matches. A similarity level of 98% or higher was chosen because intraspecific genetic variation in most fish species studied to date typically falls within the 2% threshold [13,15,20–22]. This methodology utilized the BLAST algorithm (NCBI) and BOLD Identification Engine. Following these comprehensive validations, the genetic sequences were duly submitted to the NCBI database, contributing to the broader body of scientific knowledge and facilitating further research and insights into the studied species (S2 Table).

To assess relationships between the samples and reference data, phylogenetic trees were constructed. Sequences were aligned in MEGA11 and saved as FASTA files. For COI analysis in sharks, sequences of *Rhynchobatus australiae* and *Glaucostegus granulatus* served as outgroups, while that of *R. australiae* was used for ND2. In batoids, sequences of *Chiloscyllium punctatum* and *C. griseum* were used as outgroups for COI and ND2, respectively. The model for the sequence alignment was selected using IQ-TREE ModelFinder [39]. This method, integrated within IQ-TREE V2.3.6, was employed to identify the best substitution model based on the Bayesian Information Criterion (BIC), which was TIM2 + F + I + G4. Bayesian inference of phylogeny was conducted using MrBayes (v3.2.7) [40]. A Markov Chain Monte Carlo (MCMC) analysis was executed in MrBayes with a total of 1,000,000 generations. The posterior distribution was sampled every 100 generations. Four chains were used to improve exploration of the posterior distribution. After the analysis, 25% of the samples were discarded as burn-in to eliminate the influence of the starting state. Convergence diagnostics, including the standard deviation of split frequencies and effective sample sizes (ESS), were assessed to ensure reliability of the analysis. The consensus tree was generated by Figtree V1.4.4 with posterior probabilities representing the clade support [41].

These trees offered a visual representation of the evolutionary placements for both sharks and batoids. The genetic distances were quantified using the Kimura-2-Parameter (K2P) method, gauging the variations of intra- and inter- specific diversity of species and genera of samples in this study. For this analysis, the reference sequences of unidentified species were excluded. Because several species in this study were represented by a small sample size, sequences in NCBI and BOLD databases under the same scientific names were used to enable intraspecific comparison.

## Results

There were 153 samples with successful amplification of either COI or ND2 fragments or both. The two DNA barcoding fragments offered different advantages (Table 1). COI fragments showed a better rate of amplification success than ND2, while ND2 provided more characteristics for comparison. A larger number of ND2 sequences could not be matched with any references in the database, whereas matches were found for most COI sequences; however, some matched sequences were attributed to more than one species and were therefore inconclusive. A matching result was not considered as identification; rather, conclusive species identification was based on morphological examination. In cases where only DNA sequences were available, the genetic data showed only possible identification.

With the combined use of morphological and genetic data, the samples in this study unveiled a total of 28 shark species and 32 batoid species. The process of arriving at consensus species heavily relied on morphological traits, with

**Table 1. Comparisons of COI and ND2 fragments as DNA barcoding for elasmobranch species identification from 153 samples.**

| Criteria | COI | ND2 |
| --- | --- | --- |
| Number of sequences of successful amplification | 130 | 99 |
| Length of fragments obtained (bp) | 520-676 | 970-1047 |
| Number of sequences NOT matched with any reference sequences or matched with unidentified species | 9 | 31 |
| Number of sequences matched with multiple species | 26 | 0 |

genetic data serving a supplementary role. Species identification was consistent between morphological and genetic analyses, provided that intraspecific genetic variation did not exceed 2% (Table 2; Figs 1–4). In one case, phylogenetic analysis of a sample assigned to *Orectolobus leptolineatus* on morphological criteria matched an unidentified *Orectolobus* sequence (Fig 2A).

Not all samples could be identified. Problems encountered in species identification could be grouped into four categories.

### 1. Insufficient information for comparison/ lack of species registered in the database

Specific challenges emerged in cases where information on references was unavailable within established databases. In these instances, the species identification was based primarily on morphological characteristics of the samples. These samples included rarely found species such as *Hemitrygon laosensis*, *Echinorhinus brucus*, *Squalus montalbani*, *Platyrhina psomadakisi*, *Torpedo sinuspersici*, and *Cruriraja andamanica*. Genetic data for some species such as *Rhinobatos ranongensis* were not available for public access. In these cases, we relied solely on morphology for species identification.

For *Squalus hemipinnis*, the reference sequences did not fully meet the sequence alignment criteria of this study, and also did not align with any known species. Although the phylogenetic tree placed it in close proximity to *S. brevirostris*, there was a 2.5% genetic difference. An alignment of short fragments of ND2 of our specimen and sequences in the NCBI database (529 base pairs—not shown) listed under *S. hemipinnis* revealed their similarity. This information was omitted and is not presented here. By combining morphological characteristics and partial DNA barcoding data, the consensus identification for this sample was determined to be *S. hemipinnis*.

### 2. Taxonomic revision/ redescription/ new species assignments

Taxonomic revisions present yet another challenge that impedes the comprehensive utilization of DNA barcodes in databases. Many of these barcodes were deposited prior to taxonomic revisions and have not been updated, resulting in classification of samples under different names. Such instances were observed in *Centrophorus uyato*, several species of rays in the family Dasyatidae and *Aetobatus ocellatus*; these were all matched with species bearing the names before revisions. Consensus as to species was drawn based on morphological characteristics and published distributional ranges of taxa in current literature. The names used when the samples were deposited were likely synonyms and require updating. Our DNA sequence data could be compared with sequences recently deposited, after taxonomic revision. Nevertheless, some revised taxa, namely *T. crozieri*, lacked references for their DNA barcodes. Given the unavailability of morphology data for these references, we attempted to establish DNA barcoding profiles that aligned with their corresponding morphological characteristics under the assigned names.

The samples of *Glaucostegus* in this study were previously recognized as *Glaucostegus* cf. *granulosus* because they showed morphological differences from *G. granulosus*. COI fragments from our samples did not overlap with most

**Table 2. Species identification based on morphological and genetic characteristics.**

| Morphological identification | Genetic identification based on COI and/or ND2 (best match) | Phylogenetic tree placement | Consensus species | Problem |
|---|---|---|---|---|
| **SHARKS** | | | | |
| ECHINORHINIFORMES | | | | |
| *Echinorhinus brucus* | *Echinorhinus* sp. | *Echinorhinus* sp. | *E. brucus* | Insufficient information for comparison |
| SQUALIFORMES | | | | |
| *Squalus hemipinnis* | not matched | *S. brevirostris* | *S. hemipinnis*[*1] | Insufficient information for comparison |
| *Squalus montalbani* | *Squalus* sp. | *Squalus* sp. A | *S. montalbani* | Insufficient information for comparison |
| *Centrophorus uyato* | *C. granulosus/ C. zeehaani/ C. uyato* | *Cen. uyato* | *C. uyato*[*2] | Taxonomic revision |
| *Etmopterus fusus* | not matched | not grouped with any species (COI); *E. splendidus* (ND2) | *E. fusus* | Discordance between morphology and genetics |
| ORECTOLOBIFORMES | | | | |
| *Orectolobus leptolineatus* | *O.* cf. *leptolineatus* | *O. leptolineatus* | *O. leptolineatus* | None |
| *Chiloscyllium griseum* | *C. griseum* | *C. hasselti/ C. griseum* | *C. griseum* | None |
| *Chiloscyllium hasselti* | *C. hasselti/ C. griseum* | *C. hasselti/ C. griseum* | *C. hasselti* | Cryptic morphological similarity |
| *Chiloscyllium indicum* | *C. indicum* | *C. indicum* | *C. indicum* | None |
| *Chiloscyllium punctatum* | *C. punctatum* | *C. punctatum* | *C. punctatum* | None |
| *Stegostoma tigrinum* | *S. fasciatum* | *S. fasciatum* | *S. tigrinum*[*3] | None |
| LAMNIFORMES | | | | |
| *Isurus oxyrinchus* | *I. oxyrinchus* | *I. oxyrinchus* | *I. oxyrinchus* | None |
| CARCHARHINIFORMES | | | | |
| *Atelomycterus marmoratus* | *A. marmoratus* | *A. marmoratus* | *A. marmoratus* | None |
| *Bythaelurus lutarius* | *B. hispidus* | *B. hispidus* (COI); not grouped with any species (ND2) | *B.* cf. *lutarius* | Discordance between morphology and genetics |
| *Iago mangalorensis* | *Iago* sp. | *Iago* sp. A | *I. gopalakrishnani*[*4] | Taxonomic revision |
| *Mustelus stevensi* | *Mustelus* sp./ *M. lenticulatus/ M. stevensi* | *M. lenticulatus* (COI); *M. stevensi* (ND2) | *M. stevensi* | Cryptic morphological similarity |
| *Carcharhinus amboinensis* | *C. amboinensis* | *C. amboinensis* | *C. amboinensis* | None |
| *Carcharhinus brevipinna* | *C. brevipinna* | *C. brevipinna* | *C. brevipinna* | None |
| *Carcharhinus falciformis* | *C. falciformis/ C. brevipinna* | *C. falciformis* | *C. falciformis* | Cryptic morphological similarity |
| *Carcharhinus leucas* | *C. leucas* | *C. leucas* | *C. leucas* | None |
| *Carcharhinus limbatus* | *C. limbatus* | *C. limbatus* | *C. limbatus* | None |
| *Carcharhinus longimanus* | *C. longimanus* | *C. longimanus* | *C. longimanus* | None |
| *Carcharhinus melanopterus* | *C. melanopterus* | *C. melanopterus* | *C. melanopterus* | None |
| *Carcharhinus sorrah* | *C. sorrah/ C. sealei* | *C. sorrah* | *C. sorrah* | Cryptic morphological similarity |
| *Prionace glauca* | *P. glauca* | *P. glauca* | *P. glauca* | None |
| *Galeocerdo cuvier* | *G. cuvier* | *G. cuvier* | *G. cuvier* | None |
| *Sphyrna lewini* | *S. lewini* | *S. lewini* | *S. lewini* | None |
| *Sphyrna mokarran* | *S. mokarran* | *S. mokarran* | *S. mokarran* | None |
| **BATOIDS** | | | | |
| RHINOPRISTIFORMES | | | | |
| *Rhynchobatus australiae* | *R. australiae* | *R. australiae* | *R. australiae* | None |
| *Rhinobatos ranongensis* | no matched | *Rhinobatos* cf. *borneensis* | *R. ranongensis* | Insufficient information for comparison |

*(Continued)*

**Table 2.** (Continued)

| Morphological identification | Genetic identification based on COI and/or ND2 (best match) | Phylogenetic tree placement | Consensus species | Problem |
|---|---|---|---|---|
| *Glaucostegus younholeei* | *Glaucostegus* sp. (later as *G. younholeei*)/ *G. granulosus* | *Glaucostegus* sp./ *G. granulosus* | *G. younholeei* *5 | Cryptic morphological similarity |
| *Platyrhina psomadakisi* | not matched | not grouped with any species | *P. psomadakisi* | Insufficient information for comparison |
| TORPEDINIFORMES | | | | |
| *Torpedo sinuspersici* | not matched | *Torpedo* sp./ *T. sinuspersici* | *T. sinuspersici* | Insufficient information for comparison |
| *Benthobatis moresbyi* | *B. moresbyi* | *B. moresbyi* (COI); not grouped with any species (ND2) | *B. moresbyi* | None |
| *Narcine maculata* | *N. maculata* | *N. maculata* | *N. maculata* | None |
| *Narcine prodorsalis* | *N. maculata/ Narcine* sp./ *N.* cf. *oculifer* | *Narcine* sp. | *N. prodorsalis* | Cryptic morphological similarity |
| *Narcine timlei* | *N. timlei/ N. brunnea* | *N. timlei* | *N. timlei* | Cryptic morphological similarity |
| RAJIFORMES | | | | |
| *Orbiraja powelli* | *O. powelli* | *O. powelli* | *O. powelli* | None |
| *Cruriraja andamanica* | not matched | *Cruriraja triangularis* (COI); *C. hulleyi* (ND2) | *C. andamanica* | Insufficient information for comparison |
| MYLIOBATIFORMES | | | | |
| *Hexatrygon bickelli* | *H. bickelli* | *H. bickelli* | *H. bickelli* | None |
| *Gymnura poecilura* | *G. poecilura* | *G. poecilura* | *G. poecilura* | None |
| *Brevitrygon heterura* | *B.* (*Himantura*) *walga/ B. heterura* | *B. walga/ B. heterura* | *B. heterura* *6 | Taxonomic revision |
| *Hemitrygon bennetti* | *H.* (*Dasyatis*) *bennetti* | *H. bennetti* | *H. bennetti* | None |
| *Hemitrygon laosensis* | not matched | not grouped with any species | *H. laosensis* | Insufficient information for comparison |
| *Himantula undulata* | *H. undulata* | *H. undulata* | *H. undulata* | None |
| *Maculabatis gerrardi* | *M. gerrardi/ M. macrura* | *M. macrura/ M. gerrardi* | *M. gerrardi* | Cryptic morphological similarity |
| *Maculabatis pastinacoides* | *M. pastinacoides* | *M. pastinacoides* | *M. pastinacoides* | None |
| *Neotrygon caeruleopunctata* | *N. kuhlii/ N. malaccensis/ N. caeruleopunctata* | *N. kuhlii/ N. varidens/ N. caeruleounctata* | *Neotrygon caeruleopunctata* *7 | Taxonomic revision |
| *Neotrygon varidens* | *N. trigonoides/ N. kuhlii/ N.* cf. *kuhlii* | *N. kuhlii/ N. trigonoides* | *N. varidens* *7 | Taxonomic revision |
| *Pateobatis jenkinsii* | *P.* (*Himantura*) *jenkinsii* | *P. jenkinsii* | *P. jenkinsii* | None |
| *Pateobatis uarnacoides* | *P.* (*Himantura*) *uarnacoides* | *P. uarnacoides* | *P. uarnacoides* | None |
| *Pteroplatytrygon violacea* | *P. violacea* | *P. violacea* | *P. violacea* | None |
| *Taeniura lymma* | *T. lymma* | *T. lymma* | *T. lymma* | None |
| *Telatrygon biasa* | *T.* (*Dasyatis*) *zugei/ T. biasa* | *T.* (*Dasyatis*) *zugei* (COI); *T.* (*Dasyatis*) *zugei/ T. biasa* (ND2) | *T. biasa* *8 | Taxonomic revision |
| *Telatrygon crozieri* | *T.* (*Dasyatis*) *zugei* | *T.* (*Dasyatis*) *zugei/ T. biasa* | *T. crozieri* *8 | Taxonomic revision |
| *Urogymnus lobistoma* | *U. lobistoma* | *U. lobistoma* | *U. lobistoma* | None |
| *Myliobatis hamlyni* | *M. tobijei* | *M. tobijei* | *M. hamlyni* *9 | Taxonomic revision |
| *Aetobatus ocellatus* | *A. ocellatus/ A. narinari* | *A. ocellatus* | *A. ocellatus* *10 | Taxonomic revision |
| *Rhinoptera jayakari* | *R. jayakari* | *R. jayakari* | *R. jayakari* | None |
| *Mobula tarapacana* | *M. tarapacana* | *M. tarapacana* | *M. tarapacana* | None |

*1 short ND2 fragment comparison = *S. hemipinnis*.

*2 taxonomic revision—*C. zeehaani* = syn. of *C. uyato* often misidentified with *C. granulosus* [42].

*3 listed as *Stegostoma fasciatum* in BOLD, listed as *S. fasciatum* and *S. tigrinum* in NCBI.

*(Continued)*

**Table 2.** (Continued)

*4 taxonomic revision and species resurrection [43].

*5 *Glaucostegus* sp. in GenBank described as *G. younholeei* sp. nov. in Habib and Islam [44].

*6 taxonomic revision and geographic distribution of *Brevitrygon* [38,45].

*7 taxonomic revision of *Neotrygon kuhlii* [46–48].

*8 taxonomic revision and species resurrection [34].

*9 taxonomic revision and species resurrection [49].

*10 taxonomic revision and species resurrection [50].

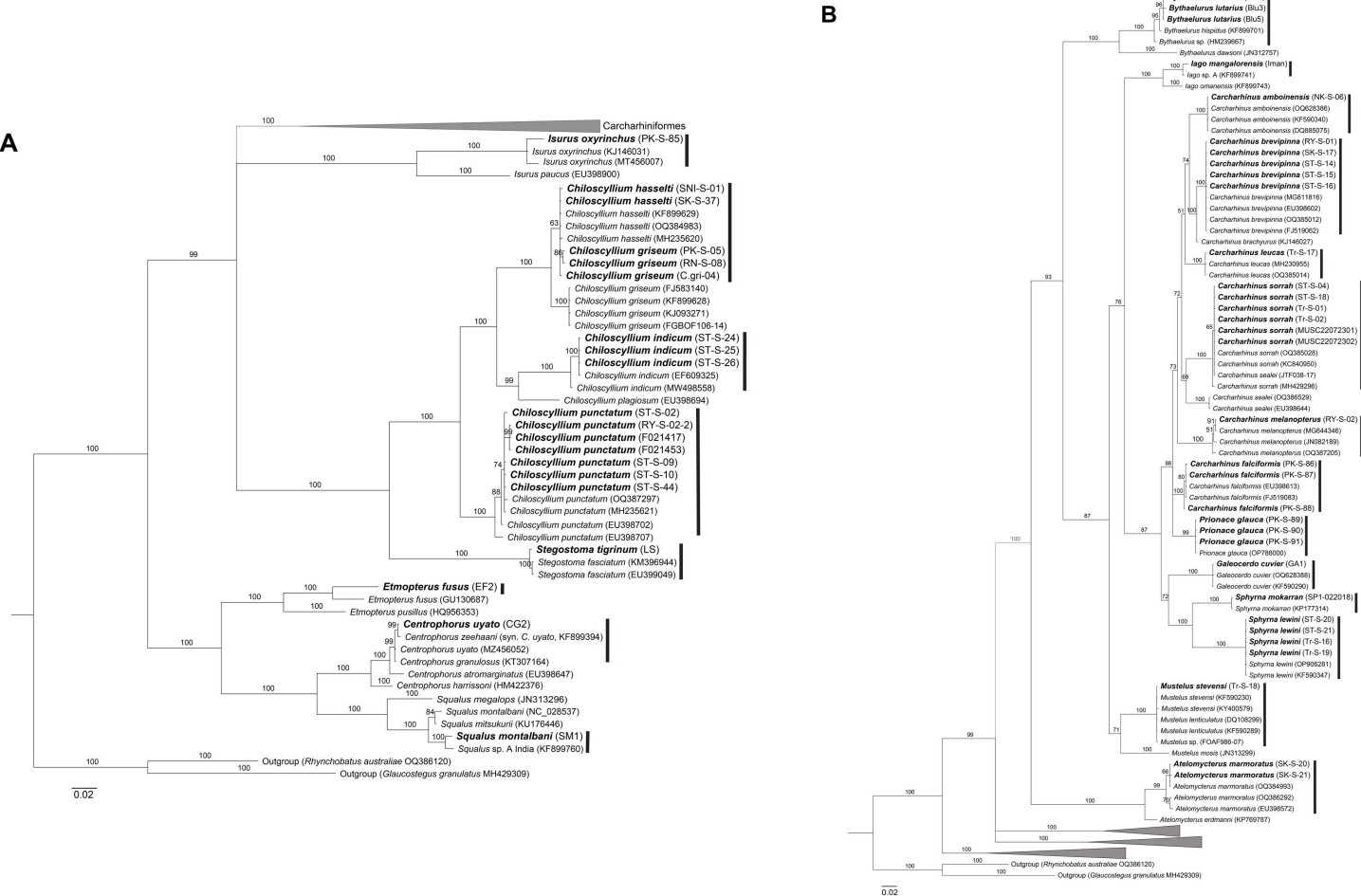

**Fig 1. Phylogenetic tree of sharks based on 654 bp COI fragments. (A)** Overall relationships of shark species analyzed in this study. **(B)** Phylogenetic tree of sharks in the order Carcharhiniformes. Bolded labels indicate samples from this study. Numbers above branches indicate bootstrap support values.

*Glaucostegus* COI sequences in the public database due to very poor DNA quality resulting in sequences that did not align well with published reference sequences, preventing the application of a 95% query coverage threshold. Instead, a minimum coverage of 60% or higher was used to achieve the best possible results. A separate tree was created for *Glaucostegus* (Fig 3C). Our DNA sequences matched an undescribed *Glaucostegus* sp. later named as a distinct species.

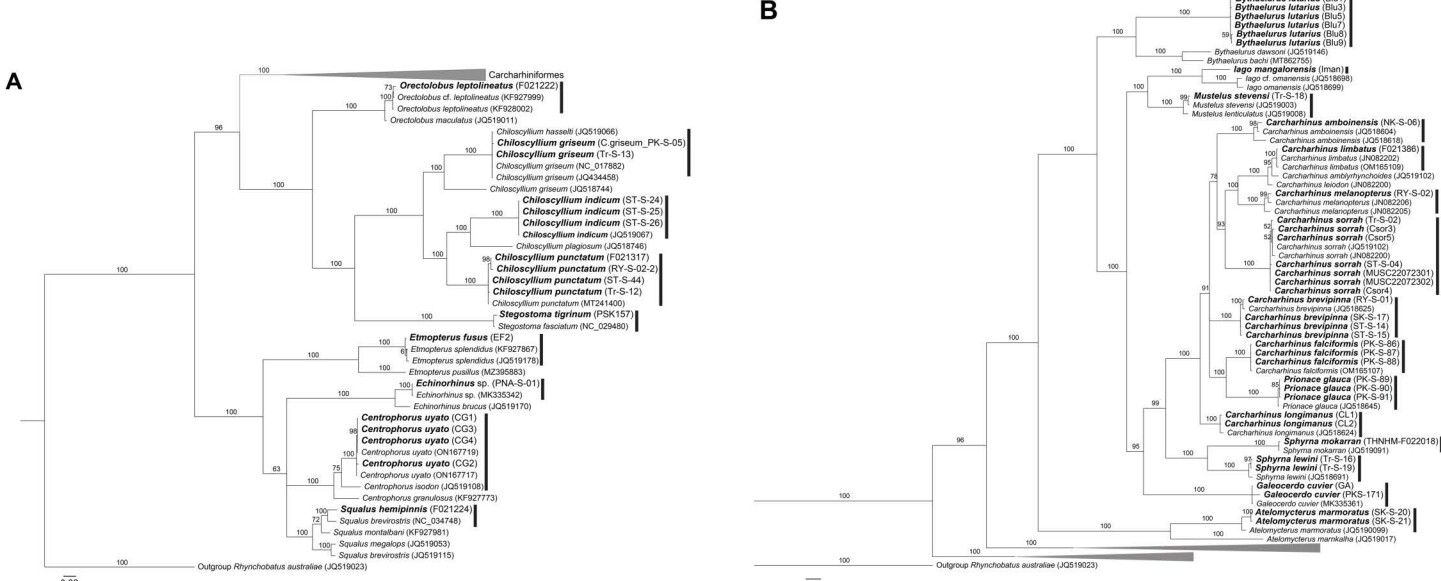

**Fig 2. Phylogenetic tree of sharks based on 1,044 bp ND2 fragments. (A)** Overall relationships of shark species analyzed in this study. **(B)** Phylogenetic tree of sharks in the order Carcharhiniformes. Bolded labels indicate samples from this study. Numbers above branches indicate bootstrap support values.

The updated publication of the sequence information in the NCBI website allowed us to place the valid name on these samples as *G. younholeei* [44].

The reassignment of species in the genus *Myliobatis* clarified the presence of two closely related species, *M. hamlyni* and *M. tubijei*. *Myliobatis hamlyni* was previously considered a junior synonym of *M. tobijei* but later confirmed as a valid species. The name applied to sequences of *M. tobijei* that matched with our sample likely followed the original designation, and was not subsequently updated. No sequences of *M. hamlyni* were available for comparison with our data. Based on morphological examination and distribution range, we conclude that the consensus species of this sample is *M. hamlyni*.

The houndshark sample was initially identified as *Iago mangalorensis* based on morphological similarities, which more closely resembled this species than other members of the genus *Iago*, prior to the formal description of the new species *I. gopalakrishnani* [43]. At the time, genetic data for *I. mangalorensis* were not available for comparison. Although our sample clustered with several undescribed *Iago* species, it was provisionally designated as *I. mangalorensis*. Following the formal description of *I. gopalakrishnani*, the sample is now considered to belong to this newly recognized species.

### 3. Cryptic morphological similarity

There were discrepancies in the labeling of sequences when comparing the NCBI and BOLD databases for a few species, such as *Chiloscyllium hasseltii*, which closely resembles *Chiloscyllium griseum*; *Carcharhinus falciformis*, misidentified as *Carcharhinus. brevipinna*; *Carcharhinus sorrah*, misidentified as *Carcharhinus seali*; and *Maculabatis gerrardi*, occasionally misidentified as *M. macrura*. These highly species pairs may be misidentified by non-specialists. For example, *Chiloscyllium hasseltii* and *Chiloscyllium griseum* are morphologically similar as adults but can be distinguished in the juvenile stage based on the presence of dark brown bands. Similarly, *M. gerrardi* and *M. macrura* both exhibit white spots on their bodies but differ in the distribution of these markings and the morphometric ratios (head length to disc width and disc length to disc width).

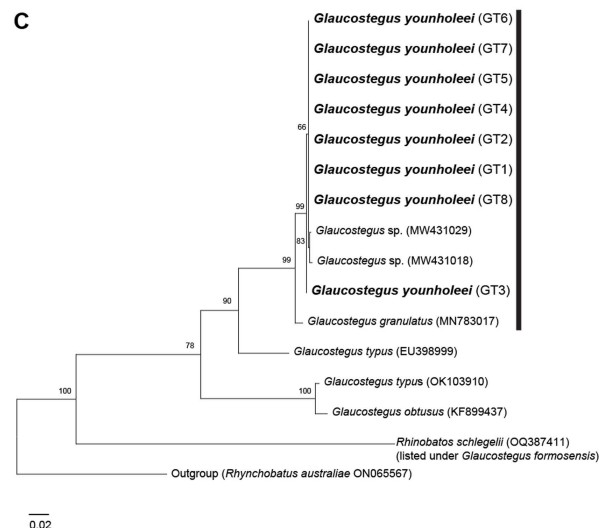

**Fig 3. Phylogenetic tree of batoids based on 652 bp COI fragments. (A)** Overall relationships of batoid species analyzed in this study. **(B)** Phylogenetic tree of batoids in the family Dasyatidae. **(C)** Phylogenetic tree of batoids in the genus *Glaucostegus*. Bolded labels indicate samples from this study. Sequences MW431029, MW431018 were identified as *G. younholeei* (Habib and Islam 2021). Numbers above branches indicate bootstrap support values.

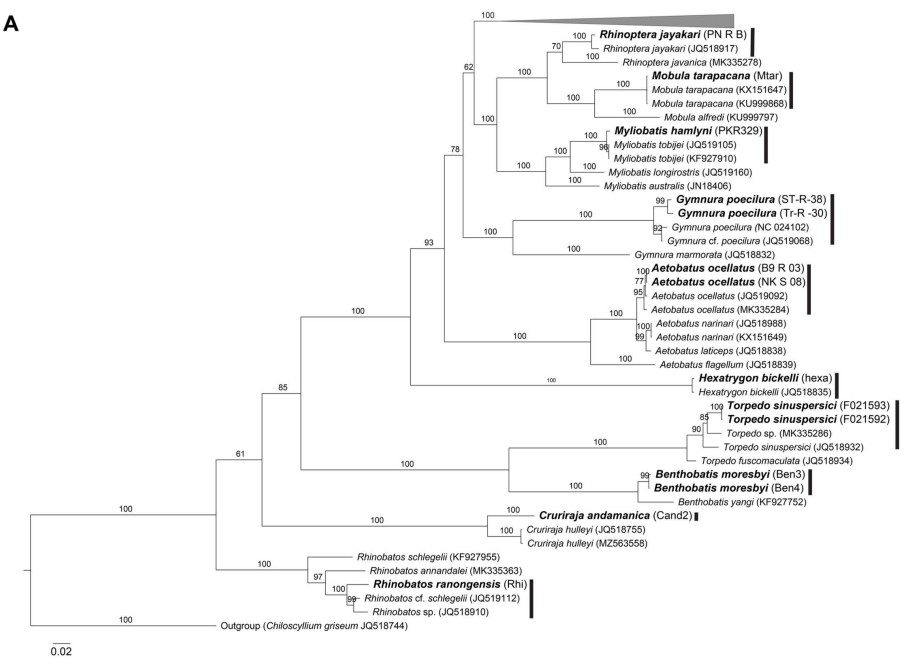

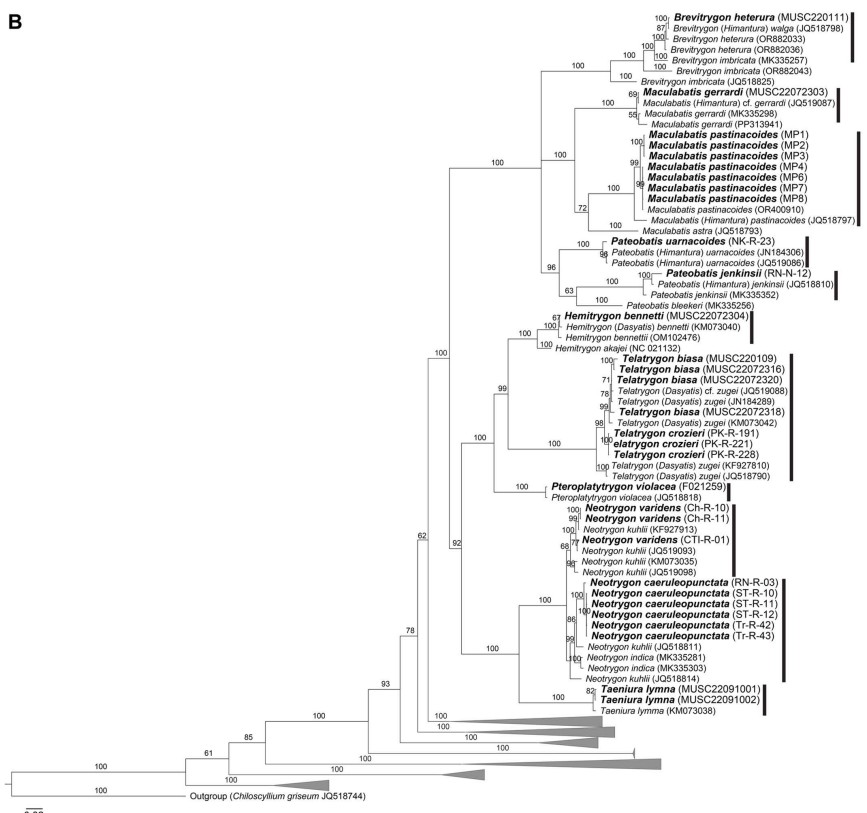

**Fig 4. Phylogenetic tree of batoids based on 1,044 bp ND2 fragments. (A)** Overall relationships of batoid species analyzed in this study. **(B)** Phylogenetic tree of batoids in the family Dasyatidae. Bolded labels indicate samples from this study. Numbers above branches indicate bootstrap support values.

Some species are unfamiliar and therefore misidentified as related species such as *Mustelus stevensi* occasionally misidentified as *Mustelus lenticulatus* and *Narcine timlei* occasionally misidentified as *N. brunnea*. These samples exhibited distinct morphological characteristics from their congeners. In such cases, we placed reliance on the morphological characteristics of these species, recognizing the potential for mislabeling in the references. Furthermore, we cross-referenced synonyms of these species with available literature, giving priority to accepted names. Consequently, the consensus species identification remained consistent with the initially assigned species name.

Sequences under name *Narcine prodorsalis* were either not present or unavailable for public access in both NCBI and BOLD. However, our *N. prodorsalis* COI sequence matched with an undescribed *Narcine* sp. and one *N. maculata*. The phylogenetic placement of our sample and the other *Narcine* species showed that the *N. maculata* that matched our sequence was likely mislabeled because all other *N. maculata* reference sequences formed a distinct clade.

## 4. Discordance between morphology and genetics

We also encountered discrepancies between morphology and genetics, as in the case of *Bythaelurus* cf. *lutarius*. This species exhibited clear morphological differences from its counterpart, *B. hispidus*, and other congeners. However, their COI sequences aligned with that of *B. hispidus* (Fig 1B) while ND2 sequences did not align with any *Bythaelurus* species (Fig 2B). These morphological distinctions included the absence of papillae on the tongue and roof of the mouth, variations in head shape and differences in the origins of the dorsal fins [5]. As a result, a consensus identification could not be definitively established, and the specimens were provisionally assigned as *B.* cf. *lutarius*.

Another challenging case was found in a sample morphologically identified as *Etmopterus fusus*. The COI sequence of this sample did not match any references in either NCBI or BOLD while its ND2 sequence matched an *E. splendidus* sample from Taiwan. Our sample from the Andaman Sea displayed clear morphological characteristics consistent with *E. fusus*, including the presence of four black bands above the anal fin continuing to the caudal fin. This feature distinguished it from *E. splendidus*, which displays a different black band pattern in the same position. The distribution ranges of these 2 species are clearly different with *E. fusus* in the Eastern Indian Ocean and northern Western Australia, and *E. splendidus* in the Northwestern Pacific Ocean. The single reference COI sequence of *E. fusus* which came from a sample in New Caledonia showed a genetic distance from our sample greater than 6% based on 659 bp. Based on the morphological characteristics and the collecting locality of our sample, the consensus identification was *E. fusus*.

The average intraspecific genetic distance for sharks and batoids across both gene fragments, COI and ND2, was about 1% (Table 3). However, the maximum intraspecific genetic distance was much greater. In sharks, the highest COI variation was between our *Etmopterus fusus* sample and the references (6.32%). A similar ND2 pattern was seen in *Echinorhinus brucus*. Among batoids, samples of *Brevitrygon imbricata* and *Aetobatus flagellum* presented the highest intraspecific genetic distances for both genes. Notably, *Himantura leoparda* and *Narcine* spp. showed COI intraspecific variation exceeding 4%. The average genetic differentiation among congeneric species exceeded 7% for both gene fragments.

**Table 3. Genetic distances (K2P percentage) based on COI and ND2 of elasmobranch species.**

| Level of genetic comparison | Taxa | COI | | | ND2 | | |
|---|---|---|---|---|---|---|---|
| | | Mean (S.E.) | Min | Max | Mean (S.E.) | Min | Max |
| Variation among individuals of the same species | Sharks | 0.92 (0.26) | 0.00 | 6.32 | 0.76 (0.21) | 0.00 | 5.37 |
| | Batoids | 0.86 (0.13) | 0.00 | 5.12 | 1.26 (0.31) | 0.00 | 10.49 |
| Variation among species of the same genus | Sharks | 7.50 (0.54) | 0.05 | 17.73 | 10.88 (0.57) | 0.39 | 21.01 |
| | Batoids | 10.42 (0.45) | 0.21 | 26.40 | 11.31 (0.59) | 0.58 | 32.38 |

However, very low interspecific genetic distances were observed between our *Etmopterus fusus* sample and the NCBI reference of *E. splendidus* and between our *Mustelus stevensi* sample and the reference *M. lenticulatus*. Among batoids, very low interspecific genetic variation was observed in *Neotrygon kuhlii* and other *Neotrygon* species and also within the genus *Telatrygon*. Similarly, low genetic variation was observed among species of *Rhinoptera*, *Aetobatus*, *Mobula*, and *Myliobatis* in the NCBI reference sequences.

The distribution of K2P genetic distances in sharks and batoids revealed distinct differences between intra- and interspecific variation (Figs 5 and 6). Over 80% of our samples fell within 2% genetic distance of their assigned species. However, interspecific genetic variation varied across both genes. Some genera, such as *Etmopterus* and *Echinorhinus*, showed highly similar sequences, likely due to limited available genetic data. In contrast, other genera, such as *Neotrygon*, *Telatrygon*, and *Aetobatus*, have undergone taxonomic revisions, and current scientific names of many sequences have not been updated.

Sequences from our samples of *Rhinobatos ranongensis*, *Cruriraja andamanica*, and *Hemitrygon laosensis*, did not match any published species since reference sequences were unavailable. These DNA sequences showed distinct genetic diversity compared to their congeneric species.

## Discussion

The implementation of DNA barcoding significantly enhances the taxonomic clarification of cartilaginous fishes, offering an alternative to the intricate and time-consuming process of species identification for non-taxonomists [15,51]. However, morphological examination still remains the key method. Among the well-recognized mitochondrial gene candidates serving as DNA barcodes for cartilaginous fishes, COI and ND2 stand out [7,12,13,15,20]. COI tends to demonstrate higher success rates in amplification compared to ND2 due to its shorter length. The design of shorter fragments within the ND2 region amplification process has bolstered its efficacy, providing additional and valuable insights for future application. It is important not to pass judgment on the superiority of one gene over the other, as each presents distinct advantages. COI has enjoyed a decade-long legacy as a DNA barcode for fishes [12,13,15,21,22], whereas the acceptance of ND2 as a DNA barcode for chondrichthyans came later [7,52]. The extensive utilization of COI is mirrored by the larger repository of COI sequences available across databases in comparison to ND2 sequences. However, ND2 sequences offer the advantage of increased length, subsequently affording more characteristics for meaningful comparison. Consequently, a synergistic employment of both genes holds promise in the accurate assessment of elasmobranch diversity. At present, the number of ND2 sequences available for comparison is fewer than COI. COI still serves as important DNA barcode but needs serious taxonomic updating.

The utility of DNA barcoding has contributed to new species records [5,22,53,54]. In this study, DNA barcoding clarified specimens initially identified as *Glaucostegus* cf. *granulatus* [5], confirming them as *Glaucostegus younholeei* [44]. Additionally, DNA barcoding aids in identifying cryptic species, leading to new species descriptions and taxonomic revisions [e.g., 55–61].

Despite the considerable promise of DNA barcodes, this study has revealed several limitations. The effectiveness of this tool is contingent upon the availability of data from verified species, which is notably absent or limited from deep-sea species. Because of the rarity of specimens, identification based on morphological criteria may be mistaken resulting in erroneous molecular sequence assignment. The description of a new species, *Glaucostegus younholeei* sp. nov. [44], provided a clear identification of samples of the guitarfish previously described as *Glaucostegus* cf. *granulatus* [5] although the reference sequences in NCBI have not been updated and currently list the species as *Glaucostegus* sp. The authors of the new species have included a reference to the description of their newly described species as an update, hence providing an official name for the taxon. In addition, the presence of synonyms for certain species, such as the zebra shark, has also presented a problem, with two names existing in the databases. Although a priority is accorded to the senior synonym, *Stegostoma tigrinum* [55], several sequences of this species are still listed as *S. fasciatum*.

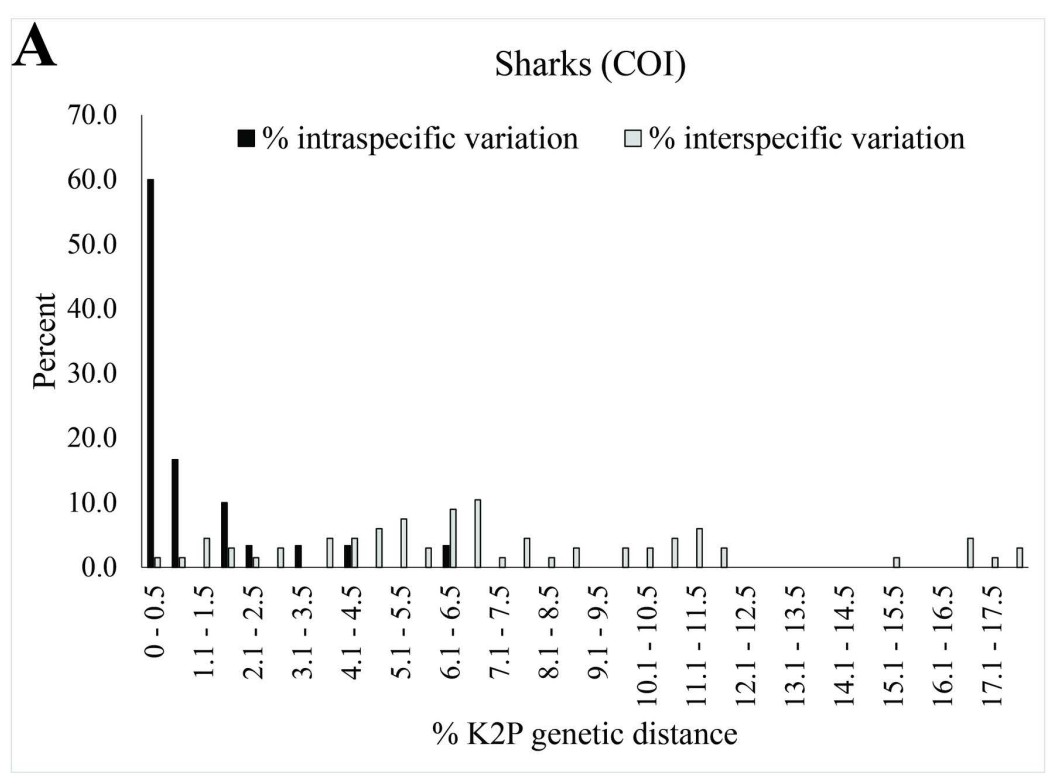

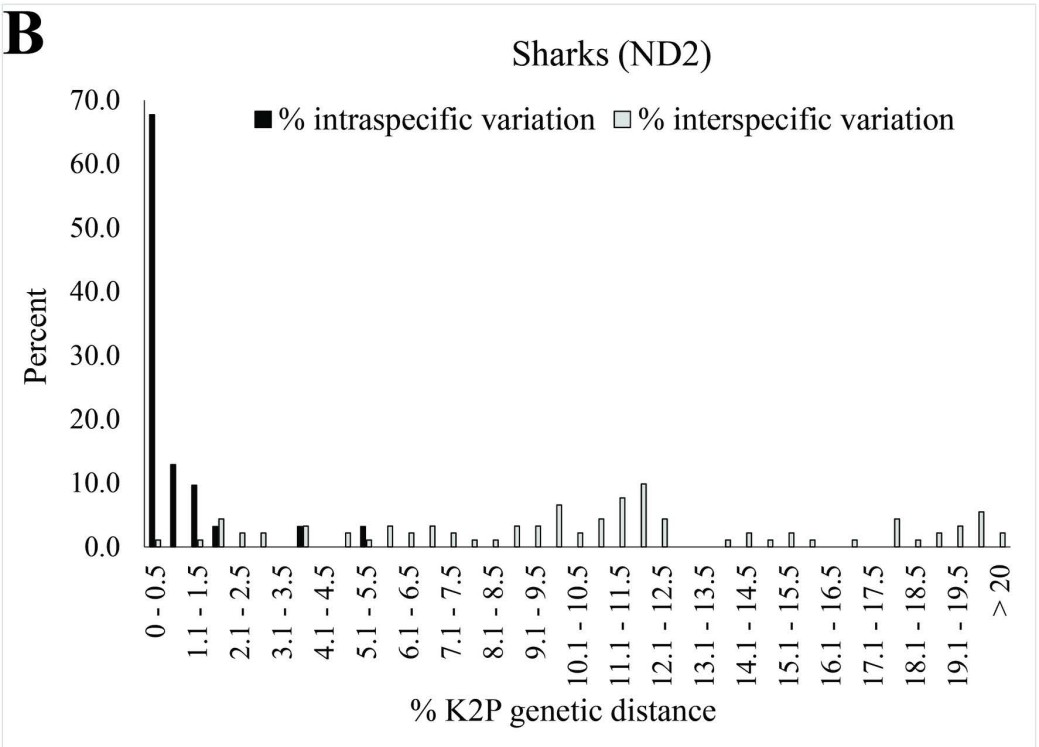

**Fig 5. Distribution of K2P genetic distances in sharks. (A)** COI fragment, with an arrow indicating *Etmopterus fusus*. **(B)** ND2 fragment, with an arrow pointing to *Echinorhinus brucus*.

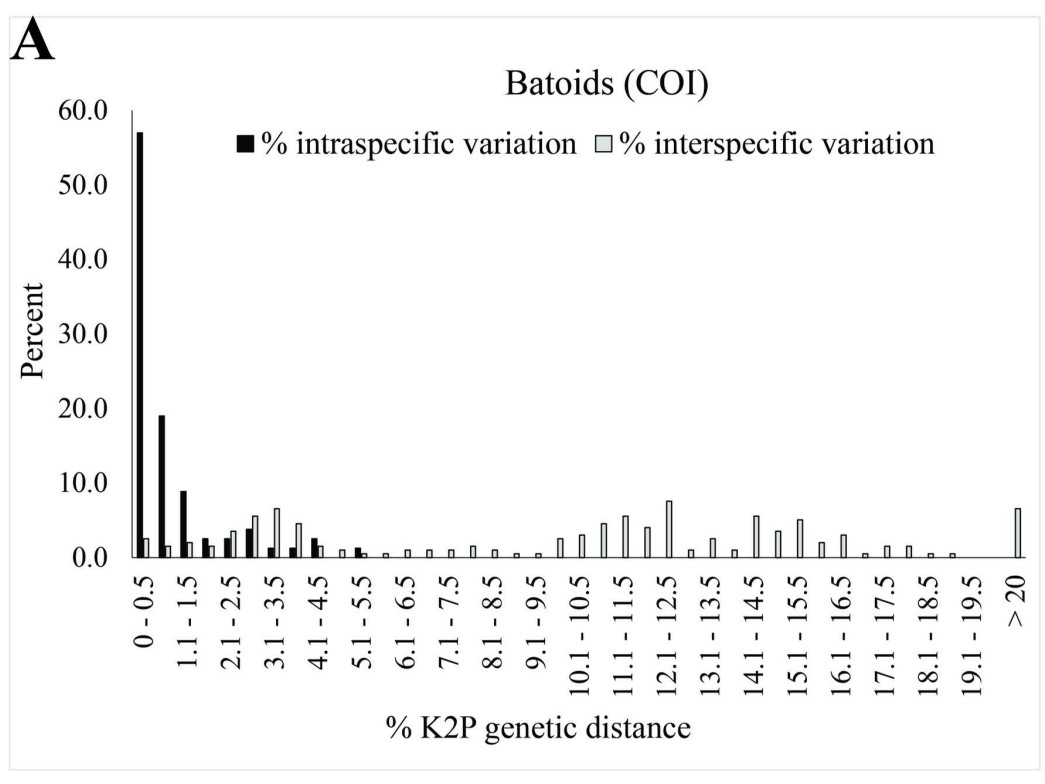

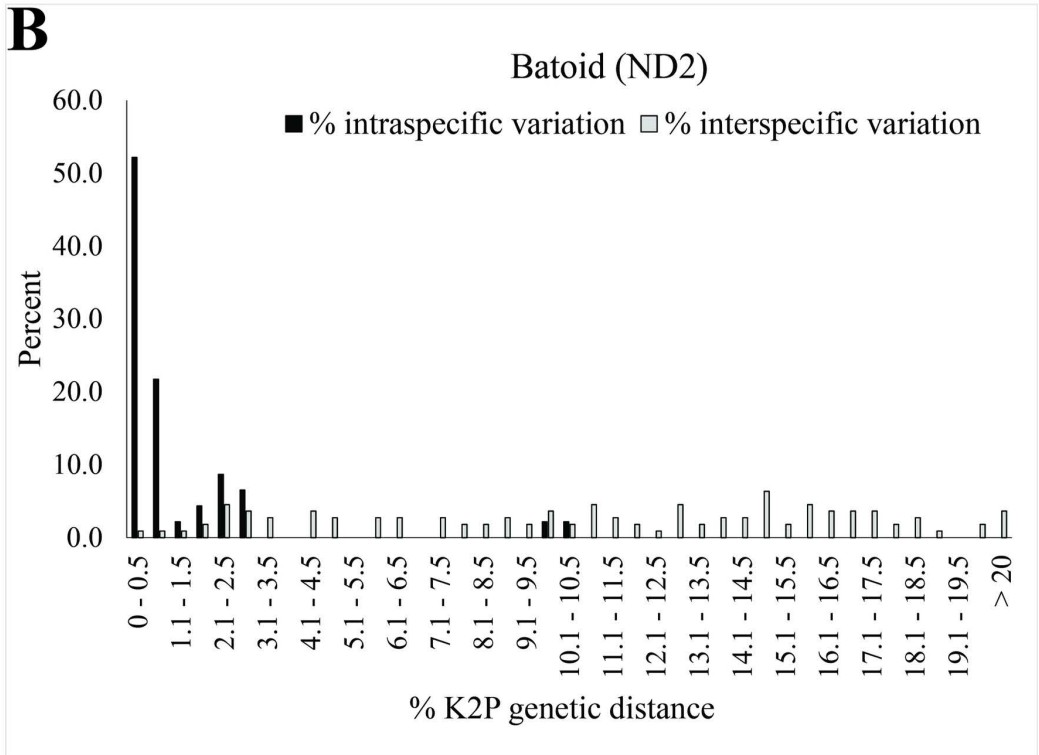

**Fig 6. Distribution of K2P genetic distances in batoids. (A)** COI fragment, with arrows indicating *Narcine timlei*, *N. brunnea*, and *Brevitrygon imbricata*. **(B)** ND2 fragment, with arrows indicating *Brevitrygon imbricata* and *Aetobatus flagellum*.

The issue of cryptic morphological similarity, due to either specimen misidentification or failure to take account of taxonomic revision, also presents a significant challenge when utilizing public databases for species verification. In such cases, it becomes necessary to rely on the morphological characteristics of specimens to confirm species assignment. The landscape of taxonomic revisions within many taxa adds complexity to the direct utilization of DNA barcodes, as species names undergo alterations over time. For example, DNA barcodes of samples identified as *Centrophorus uyato* have been matched with different *Centrophorus* species due to recent revisions that revealed *C. granulosus* as a species complex [42,56]. As a result, the COI sequences of several species including *C. uyato* were erroneously labeled as *C. granulosus*. This problem is also observed in the *Maculabatis gerrardi* complex [57]. In contrast, information on ND2 sequences is generally more recent and up-to-date, thus serving as a better reference than COI. A similar situation was found in *Aetobatus ocellatus,* which was previously grouped within the *A. narinari* species complex but later recognized as a valid species based on molecular evidence [50,58]. Sequences of samples labeled as *A. narinari* prior to this taxonomic revision could actually represent *A. ocellatus*, emphasizing the need for caution when using such references.

The extensive revision of stingrays of the family Dasyatidae has brought about significant changes in generic names and the erection of new species exemplified by *Brevitrygon* [45], *Neotrygon* [47], and *Telatrygon* [59]. The species belonging to these genera exhibit distinct morphological characteristics that serve as a primary means of identification, albeit requiring the expertise of specialists for their precise differentiation. To accelerate the identification process, DNA profiles of these species should be established. This study demonstrated such genetic distinctions through DNA barcoding, which contributes to the improved assessment of biodiversity.

Several sharks and rays are still subject to taxonomic revision due to their morphological similarity and very wide distribution ranges. The *Squalus* species complex presents wide distribution ranges and similar morphological characteristics among congeners [60–62]. Some species are thought to be endemic to certain geographic locations, but their distribution is still unclear and requires further examination, which may result in further taxonomic revision [36]. Our *Squalus* samples provisionally identified as *S. hemipinnis* and *S. montalbani*, but which did not match the reference sequences given for those taxa may in fact be distinct lineages.

Discordance between morphology and the DNA barcode assigned to *Bythaelurus* cf. *lutarius* provided difficulty in reaching consensus identification. Identification based on morphological characteristics revealed that all examined samples collected from the Thai Andaman Sea were of *B. lutarius* [5]. This species, first described as *Halaelurus lutarius* [63], is clearly distinguishable from similar looking species in the same genus [64–69]. The lack of COI sequences of *B. lutarius* from the type locality and the unavailability of the ND2 sequence of *B. hispidus*, despite being previously studied [67], poses a challenge to the use of DNA barcoding for species confirmation. Because all sampled *B.* cf. *lutarius* formed a monophyletic cluster based on longer ND2 fragments, clearly distinct from other catsharks whose sequences were available for comparison, they likely all represented the same species. Although the samples show clear morphological distinction, their location falls outside the known distribution range of *B. lutarius* and their genetic affinity within this genus remains uncertain; thus, the samples are tentatively assigned as *B.* cf. *lutarius*.

It has been long recognized that intraspecific genetic variation among metazoans generally falls within 2% [13,15,20–22]. However, low genetic variation among distinct but congeneric lineages is evident in *Mustelus* species [70]. The clear genetic differentiation between *M. stevensi* and *M. lenticulatus* [10] enabled accurate consensus identification. In contrast, certain species, such as *Etmopterus fusus*, *Squalus montalbani*, *Himantura leoparda*, *Maculabatis gerrardi*, and *Narcine timlei*, display high genetic variation within species. This may arise from either cryptic diversity or misidentification of reference samples, as seen in the species complex of *Himantura uarnak* [71], *Maculabatis gerrardi* [57], and *Narcine* species [72]. Therefore, while a 2% genetic difference within species is helpful, it cannot serve as an absolute criterion for species delimitation. Comparisons of genetic differentiation based on COI and ND2 fragments showed slightly greater interspecific differences in ND2 compared to COI possibly due to the utilization of longer fragments in the analyses. Nevertheless, low genetic differentiation among distinct congeneric species may confound their recognition. For

example, our *Etmopterus fusus* sample showed high COI sequence variation compared to reference *E. fusus* but was similar to a ND2 sequence of *E. splendidus*. Although both species are closely related and rarely available for thorough examinations [73], clear difference in their geographical distribution [74] helps identify our sample as *E. fusus*. While the influx of DNA barcoding sequences is expected to continue, its application should be approached with caution. Given the dearth of expert chondrichthyan taxonomists at both local and global scales, the utilization of molecular markers is important for facilitating biodiversity assessment [75]. The data gathered by our present study both contribute to the sample of genetic information from better-known species and provide new genetic information of several species not available in public databases. Despite this great benefit, the clarity of the species concept is crucial, and traditional taxonomic identification methods should not be disregarded [76].

Discordance between morphological and genetic characteristics and misidentification of closely related species may result from hybridization, which has been reported many times in elasmobranchs [77–82]. In such cases, DNA barcoding cannot resolve taxonomic identification and necessitates assessment of nuclear DNA. While hybridization and introgression have been documented in a few elasmobranch groups, many remain unexplored. Given that environmental changes may influence reproductive behaviors of these fishes, baseline data on genetics across different species are crucial for understanding these processes [80].

As elasmobranchs serve as resources for both local consumption and international export, prohibition of the use of these resources is impractical. On the other hand, regulation of fisheries based on solid scientific foundation and correct species assignment will help sustain long-term exploitation of these resources. To achieve this level of regulation, fundamental knowledge on species and genetic diversity is important and requires thorough examination, especially of local populations [18]. DNA barcoding data then serve as references for the authentication of meat products [14,83]. However, since taxonomic revisions are ongoing and updates may not always keep pace, maintaining well-curated voucher libraries and local reference collections is essential for accurate species verification. Limited data on biodiversity jeopardize populations of these fishes, especially those commonly used in the preparation of luxury dishes and pharmaceutical products, which usually come from non-specified species, some of which are globally vulnerable [84–88].

## Supporting information

**S1 Table. List of internal primers used in the study.**
(PDF)

**S2 Table. GenBank accession numbers of samples.** Genetic comparisons of the samples in this study were based on COI and ND2 gene fragments, using reference sequences from the NCBI and BOLD databases. Phylogenetic tree analyses were performed using the Maximum Likelihood method. Initial species identifications, based on morphological characteristics, are not presented. **Abbreviations:** GOT = Gulf of Thailand; AN = Andaman Sea; U = unable to amplify sequences; NM = sequence did not match with any data; NF = species not found in the database.
(PDF)

## Acknowledgments

We would like to thank our assistants, P. Chanmuang, T. Meejan, P. Laongdee, K. Panpong, A. Pankhort, and P. Wongmanee, who contributed to sample collection and laboratory work. The authors also thank the following organizations and individuals for providing the specimens: the Marine Fisheries Research and Development Division and Inland Fisheries Research Development Division of the Department of Fisheries, Thailand National Science Museum, Department of Marine and Coastal Resources, C. Vidthayanon, and T. Vasinopas. Language editing was assisted by P. D. Round.

## Author contributions

**Conceptualization:** Jenjit Khudamrongsawat, Tassapon Krajangdara, Thadsin Panithanarak, Ratima Karuwancharoen, Wanlada Klangnurak, Wansuk Senanan.

**Data curation:** Jenjit Khudamrongsawat, Tassapon Krajangdara, Thadsin Panithanarak, Ratima Karuwancharoen, Wanlada Klangnurak, Wansuk Senanan.

**Formal analysis:** Jenjit Khudamrongsawat, Tassapon Krajangdara, Thadsin Panithanarak, Ratima Karuwancharoen, Wanlada Klangnurak, Pattarapon Promnun.

**Funding acquisition:** Tassapon Krajangdara, Wansuk Senanan.

**Investigation:** Jenjit Khudamrongsawat, Tassapon Krajangdara, Thadsin Panithanarak, Wanlada Klangnurak, Pattarapon Promnun, Wansuk Senanan.

**Methodology:** Jenjit Khudamrongsawat, Tassapon Krajangdara, Ratima Karuwancharoen, Wanlada Klangnurak, Pattarapon Promnun, Wansuk Senanan.

**Project administration:** Wansuk Senanan.

**Validation:** Jenjit Khudamrongsawat, Tassapon Krajangdara.

**Writing – original draft:** Jenjit Khudamrongsawat, Tassapon Krajangdara.

**Writing – review & editing:** Jenjit Khudamrongsawat, Tassapon Krajangdara, Thadsin Panithanarak, Ratima Karuwancharoen, Wanlada Klangnurak, Pattarapon Promnun, Wansuk Senanan.

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
