## [Decision Letter · Decision Letter 0]

26 May 2025

Dear Dr. Senanan,

Thank you for submitting your manuscript to PLOS ONE. After careful consideration, we feel that it has merit but does not fully meet PLOS ONE’s publication criteria as it currently stands. Therefore, we invite you to submit a revised version of the manuscript that addresses the points raised during the review process.

We look forward to receiving your revised manuscript.

Kind regards,

Joel Harrison Gayford

Academic Editor

PLOS ONE

 [This project is funded by National Research Council of Thailand (NRCT-N25A650485).]. 

Additional Editor Comments (if provided):

Reviewers' comments:

Reviewer's Responses to Questions

**Comments to the Author**

1. Is the manuscript technically sound, and do the data support the conclusions?

Reviewer #1: Partly

Reviewer #2: Partly

Reviewer #3: Yes

2. Has the statistical analysis been performed appropriately and rigorously?

Reviewer #1: N/A

Reviewer #2: Yes

Reviewer #3: Yes

3. Have the authors made all data underlying the findings in their manuscript fully available?

Reviewer #1: No

Reviewer #2: No

Reviewer #3: Yes

4. Is the manuscript presented in an intelligible fashion and written in standard English?

Reviewer #1: Yes

Reviewer #2: No

Reviewer #3: Yes

Reviewer #1: The manuscript is generally worth publishing but there are many issues that need to be addressed before the manuscript can be accepted for publication, particularly concerning the references:

1) According to the data presented, the paper solely presents genetic data. However, in the abstract it is mentioned that there were conflicts between morphology and genetics, and it is mentioned in several instances that specimens were morphologically examined. I totally agree that the examination of the morphology of specimens is, of course, encouraged. Nevertheless, if the morphology was indeed examined, it needs to be explained how it was examined and the morphological data should be presented in the manuscript. This is also important for Table 2, where morphological identification is mentioned but neither has it been explained what was examined nor has the data been presented elsewhere in the manuscript.

2) Page 2, sentence “Available genetic information, though scant, poses many questions regarding taxonomic diversity in which the high levels of genetic differentiation that could not be ignored [11]”: This message was not published in the cited paper but in the reply to that paper. Therefore, the reference needs to be changed to the following one:

Weigmann 2017: Reply to Borsa (2017): Comment on ‘Annotated checklist of the living sharks, batoids and chimaeras (Chondrichthyes) of the world, with a focus on biogeographical diversity by Weigmann (2016)’. Journal of Fish Biology, 90(4), 1176–1181.

3) Table 2: The correct name of the order is “CARCHARHINIFORMES”

4) Page 7, Iago mangalorensis: The specimens should also be compared with the very recently described Iago gopalakrishnani.

5) Page 9, *2: Reference 42 is the paper currently listed as reference 43 in the reference section.

6) Page 9, *4: Reference 43 is the paper currently listed as reference 42 in the reference section.

7) Page 9, *5: Wrong paper cited, a taxonomic and biogeographic revision of Brevitrygon was carried out in the following paper:

Last, P. R., Weigmann, S., & Naylor, G. J. P. (2023). The Indo-Pacific Stingray Genus Brevitrygon (Myliobatiformes: Dasyatidae): Clarification of Historical Names and Description of a New Species, B. manjajiae sp. nov., from the Western Indian Ocean. Diversity, 15(12), 1213. https://doi.org/10.3390/d15121213

8) Page 9, *6: A taxonomic revision of Neotrygon kuhlii was not carried out in the cited references but in the following reference: Last, P.R. & White, W.T. & Séret, B. (2016) Taxonomic status of maskrays of the Neotrygon kuhlii species complex (Myliobatoidei: Dasyatidae) with the description of three new species from the Indo-West Pacific. Zootaxa, 4083(4), 533–561

9) Page 9, *7-9: Write “species resurrection” instead of “new species erection”.

10) Page 10, reference 43: see my comment 6 (regarding Glaucostegus younholeei); same applies to mentioning of this reference on page 13.

11) Page 13, first sentence of Discussion: This sentence gives the impression that barcoding can be used without making any morphological examinations. This is a very dangerous way as many examples from the literature have shown that solely relying on genetics can lead to clearly incorrect conclusions and even incorrect new species descriptions. One example for such wrong conclusions the following paper (see their Etmopterus lucifer in the order Carcharhiniformes):

Vélez-Zuazo X, Agnarsson I. Shark tales: a molecular species-level phylogeny of sharks (Selachimorpha, Chondrichthyes). Mol Phylogenet Evol. 2011 Feb;58(2):207-17. doi: 10.1016/j.ympev.2010.11.018.

The importance of integrating morphological and molecular taxonomies has been summarized, e.g., in the previously mentioned paper by Weigmann (2017) (see my comment 2 for the full citation of this paper).

12) Page 14: Reference 59 must be reference 52.

13) Page 14: Reference 42 must be reference 43 and reference 52 must be reference 53.

14) Page 14: Reference 53 must be reference 54.

15) Page 14: Reference 54 must be reference 55 (revision of Brevitrygon).

16) Page 14: Reference 11 is incorrect for revision of Neotrygon, it must be the following paper:

Last, P.R. & White, W.T. & Séret, B. (2016) Taxonomic status of maskrays of the Neotrygon kuhlii species complex (Myliobatoidei: Dasyatidae) with the description of three new species from the Indo-West Pacific. Zootaxa, 4083(4), 533–561

17) Page 14: Reference 55 must be reference 56.

18) Page 14: Reference 56-58 must be references 57-59.

19) Pages 14-15: You state that B. hispidus, B. stewarti, B. naylori, B. dawsoni, and B. canescens are found in the same region (Thai Andaman Sea). However, only B. hispidus has been found in the Andaman Sea and the paper cited for B. hispidus (reference 60) is about specimens collected from off Southwest India, not the Andaman Sea. Specimens of B. hispidus from the Andaman Sea were described in the following references:

Alcock, A. (1891) Class Pisces. In: II.–Natural history notes from H.M. Indian marine survey steamer ‘Investigator’, Commander R.F. Hoskyn, R.N. commanding.–Series II., No. 1. On the results of deep-sea dredging during the season 1890–91. The Annals and Magazine of Natural History, 8 (6), 16–34.

Springer, S. & D’Aubrey, J.D. (1972) Two New Scyliorhinid Sharks from the East Coast of Africa with Notes on Related Species. Oceanographic Research Institute, Investigational Report, 29, 1–19.

Springer, S. (1979) A revision of the catsharks, family Scyliorhinidae. NOAA Technical Report NMFS Circular, 422, 1–152.

Séret, B. (1987) Halaelurus clevai, sp. n., a new species of catshark (Scyliorhinidae) from off Madagascar, with remarks on the taxonomic status of the genera Halaelurus Gill & Galeus Rafinesque. J.L.B. Smith Institute of Ichthyology Special Publication, 44, 1–27.

Kaschner, C.J., Weigmann, S. & Thiel, R. (2015) Bythaelurus tenuicephalus n. sp., a new deep-water catshark (Carcharhiniformes, Scyliorhinidae) from the western Indian Ocean. Zootaxa, 4013 (1), 120–138. http://dx.doi.org/10.11646/zootaxa.4013.1.9

Weigmann S, Ebert DA, Clerkin PJ, Stehmann MF, Naylor GJ. Bythaelurus bachi n. sp., a new deep-water catshark (Carcharhiniformes, Scyliorhinidae) from the southwestern Indian Ocean, with a review of Bythaelurus species and a key to their identification. Zootaxa. 2016; 4208(5): 401–432. https://doi.org/10.11646/zootaxa.4208.5.1

Weigmann S, Kaschner CJ. Bythaelurus vivaldii, a new deep-water catshark (Carcharhiniformes, Scyliorhinidae) from the northwestern Indian Ocean off Somalia. Zootaxa. 2017; 4263(1): 97–119. https://doi.org/10.11646/zootaxa.4263.1.4

Weigmann S, Kaschner CJ, Thiel R. A new microendemic species of the deep-water catshark genus Bythaelurus (Carcharhiniformes, Pentanchidae) from the northwestern Indian Ocean, with investigations of its feeding ecology, generic review and identification key. PLoS One. 2018;13(12):e0207887.

20) The next sentence is not correct, either, as ND2 sequences for B. hispidus were published in the following paper:

Weigmann S, Ebert DA, Clerkin PJ, Stehmann MF, Naylor GJ. Bythaelurus bachi n. sp., a new deep-water catshark (Carcharhiniformes, Scyliorhinidae) from the southwestern Indian Ocean, with a review of Bythaelurus species and a key to their identification. Zootaxa. 2016; 4208(5): 401–432. https://doi.org/10.11646/zootaxa.4208.5.1

21) Tentative identification should be indicated by writing B. cf. lutarius (with cf. not written in italics) and likewise also for other spp. with tentative identification.

Reviewer #2: The manuscript (PONE-D-25-18018), titled “DNA barcoding for elasmobranch diversity assessment in Thailand: its advantages and limitations” is an interesting work. However, the revisions are necessary to improve the clarity, depth, and contextualization of findings. Detailed comments and suggestions are provided below, organized by section.

Adding line numbers would help reviewers provide more precise feedback.

Abstract

"mean for species identification" should be "means of species identification."

"exercise cautions" should be "exercise caution."

Suggest professional English editing for clarity and flow.

Methodological Clarifications

• Mention the primers used in this study

• DNA quantification method: Mention how DNA concentration was measured (e.g., Nanodrop, Qubit?).

• Threshold justification: Explain why 98% identity and 95% coverage were chosen as cutoffs (e.g., precedent in literature).

• Glaucostegus exception: The lowering of query coverage to 60% should be explained more clearly is this due to database fragment length?

• Model selection: The model used for phylogenetics (TIM2+F+I+G4) is specified, but the alignment software used to generate sequence alignments is not mentioned.

• Outgroups: No mention of which outgroups were used (if any) in tree construction — important for rooting and interpretation.

• Voucher information: Though storage locations are mentioned, include how voucher IDs were assigned and whether they are linked to GenBank entries.

The study relies heavily on publicly available databases (BOLD, NCBI), many of which contain mislabeled or outdated entries. This issue is discussed,

but the authors might consider emphasizing the importance of curating verified local reference

libraries.

The Results and Discussion

Consider separating detailed species-level data into Supplementary Materials.

Use summary tables/figures for clarity (e.g., show cases of taxonomic conflict or cryptic species with quick-reference tables).

Reviewer #3: 4. You need to standardize the way you write species names.

5. Clarity is needed on some inconsistencies or impossibilities of the two markers to discriminate the species. It is important to make clear the methods used to validate the information.

6. Include a brief discussion regarding the high percentage of divergence within the species Himantura leoparda and Narcine spp.

7. Phylogenetic trees need to be reconstructed or better edited, they are not clear and the species evidenced in the study, especially the groups, need to have some signaling.

Considering all the aspects raised and after a review, the article can be considered for acceptance in this journal, so I recommend a minor review.

**Do you want your identity to be public for this peer review?** For information about this choice, including consent withdrawal, please see our Privacy Policy

Reviewer #1: No

Reviewer #2: No

Reviewer #3: **Yes: ** Valdemiro Muhala

---

## [Author Response · Author response to Decision Letter 1]

14 Jul 2025

We make the corrections in response to reviewers' suggestions and questions (see attached response to reviewers for detailed revisions). In addition, we made some small adjustment to the content:

• Adding the section “Financial disclosure”

• Move the ethic statements (previously in the Acknowledgements) to method section

• Separate supporting information into 2 files: S1—primers and protocol used, S2—sample identification including morphological and genetic examination of each sample.

---

## [Decision Letter · Decision Letter 1]

4 Aug 2025

Dear Dr. Senanan,

Thank you for submitting your manuscript to PLOS ONE. After careful consideration, we feel that it has merit but does not fully meet PLOS ONE’s publication criteria as it currently stands. Therefore, we invite you to submit a revised version of the manuscript that addresses the points raised during the review process.

We look forward to receiving your revised manuscript.

Kind regards,

Joel Harrison Gayford

Academic Editor

PLOS ONE

Journal Requirements:

**Additional Editor Comments:**

After taking into account these minor revisions, I expect to be able to accept the paper for publication.

Reviewers' comments:

Reviewer's Responses to Questions

**Comments to the Author**

Reviewer #1: All comments have been addressed

Reviewer #2: (No Response)

Reviewer #3: All comments have been addressed

2. Is the manuscript technically sound, and do the data support the conclusions?

Reviewer #1: Yes

Reviewer #2: Partly

Reviewer #3: Yes

3. Has the statistical analysis been performed appropriately and rigorously?

Reviewer #1: Yes

Reviewer #2: Yes

Reviewer #3: Yes

4. Have the authors made all data underlying the findings in their manuscript fully available?

Reviewer #1: Yes

Reviewer #2: Yes

Reviewer #3: Yes

5. Is the manuscript presented in an intelligible fashion and written in standard English?

Reviewer #1: Yes

Reviewer #2: No

Reviewer #3: Yes

Reviewer #1: Thank you for revising the manuscript addressing all the reviewers' comments. The manuscript can now be accepted for publication in my opinion.

Reviewer #2: The manuscript PONE-D-25-18018R1, entitled “DNA barcoding for elasmobranch diversity assessment in Thailand: its advantages and limitations,” is interesting; however, there are instances of redundancy, and some sentences could be rephrased for improved clarity and conciseness. Additionally, the absence of line numbers makes precise referencing challenging. Including line numbers would facilitate easier review and future revisions. Please ensure consistent formatting of species names (e.g., full genus names vs. abbreviations), particularly in the Results and Discussion sections.

Abstract

"mean for species identification" should be "means of species identification."

"exercise cautions" should be "exercise caution."

Introduction

"Available genetic information, though scant, poses many questions regarding taxonomic diversity in which the high levels of genetic differentiation that could not be ignored [11]": The phrasing "in which the high levels of genetic differentiation that could not be ignored" is a bit clunky. Consider rephrasing for smoother reading.

Methods

• Did the authors perform sequence analysis? Which software did the authors use? Mention it in the ms!

Results

• The authors' decision to keep Table 2 in the Results section as it illustrates and summarizes their findings is valid. The detailed information being in the supplementary section is also appropriate.

• The authors' preference for spelling out full generic names in the text to avoid confusion, especially given the similar initial letters of many elasmobranch genera, is a valid point. However, if the authors choose to abbreviate, ensure consistency (e.g., first use full name, then abbreviate).

• addition of morphometric ratios to differentiate M. gerrardi and M. macrura provides more robust identification criteria.

• "did not aligned": This should be "did not align."

Reviewer #3: Congratulations on following the corrections left. The work is better. One thing could be added, but it doesn't take away any merit from this version. Congratulations.

**Do you want your identity to be public for this peer review?** For information about this choice, including consent withdrawal, please see our Privacy Policy

Reviewer #1: No

Reviewer #2: No

Reviewer #3: **Yes: ** Valdemiro Muhala

---

## [Author Response · Author response to Decision Letter 2]

26 Sep 2025

This is a third revision. We addressed all comments and added the line numbers to facilitate the review. Please see an attached file on the responses.

---

## [Editor Report · Decision Letter 2]

30 Sep 2025

DNA barcoding for elasmobranch diversity assessment in Thailand: its advantages and limitations

PONE-D-25-18018R2

Dear Dr. Senanan,

We’re pleased to inform you that your manuscript has been judged scientifically suitable for publication and will be formally accepted for publication once it meets all outstanding technical requirements.

Kind regards,

Joel Harrison Gayford

Academic Editor

PLOS ONE
---

## [Editor Report · Acceptance letter]

PONE-D-25-18018R2

PLOS ONE

Dear Dr. Senanan,

I'm pleased to inform you that your manuscript has been deemed suitable for publication in PLOS ONE. Congratulations! Your manuscript is now being handed over to our production team.

Kind regards,

on behalf of

Mr. Joel Harrison Gayford

Academic Editor

PLOS ONE